# Evaluation of Time and Frequency Condition Indicators from Vibration Signals for Crack Detection in Railway Axles

**Réne-Vinicio Sánchez** [1,*,†], **Pablo Lucero** [1,†], **Jean-Carlo Macancela** [1,†],
**Higinio Rubio Alonso** [2,†], **Mariela Cerrada** [1,†], **Diego Cabrera** [1,†] and **Cristina Castejón** [2,†]

1   GIDTEC Research Group, Universidad Politécnica Salesiana, Cuenca 010105, Ecuador;
    pluceroo@est.ups.edu.ec (P.L.); jmacancelap@est.ups.edu.ec (J.-C.M.); mcerrada@ups.edu.ec (M.C.);
    dcabrera@ups.edu.ec (D.C.)
2   MAQLAB Research Group, Department of Mechanical Engineering, Universidad Carlos III de Madrid,
    28911 Madrid, Spain; hrubio@ing.uc3m.es (H.R.A.); castejon@ing.uc3m.es (C.C.)
*   Correspondence: rsanchezl@ups.edu.ec
†   These authors contributed equally to this work.

**Abstract:** Railway safety is a matter of importance as a single failure can involve risks associated with economic and human losses. The early fault detection in railway axles and other railway parts represents a broad field of research that is currently under study. In the present work, the problem of the early crack detection in railway axles is addressed through condition-based monitoring, with the evaluation of several condition indicators of vibration signals on time and frequency domains. To achieve this goal, we applied two different approaches: in the first approach, we evaluate only the vibrations signals captured by accelerometers placed along the longitudinal direction and, in the second approach, a data fusion technique at the condition indicator level was conducted, evaluating six accelerometers by merging the indicator conditions according to the sensor placement. In both cases, a total of 54 condition indicators per vibration signal was calculated and selecting the best features by applying the Mean Decrease Accuracy method of Random Forest. Finally, we test the best indicators with a K-Nearest Neighbor classifier. For the data collection, a real bogie test bench has been used to simulate crack faults on the railway axles, and vibration signals from both the left and right sides of the axle were measured. The results not only show the performance of condition indicators in different domains, but also show that the fusion of condition indicators works well together to detect a crack fault in railway axles.

**Keywords:** railway axles; crack detection; condition monitoring; time-domain features; frequency-domain features; random forest classifier; feature extraction; feature selection

## 1. Introduction

The railway transportation has a rapid growth worldwide, railway safety is a subject of high interest in the research field. Railway axles are one of the most critical elements in railway transportation systems, and failures such as a cracked axle can lead to the derailment and probably human and economic losses. Therefore, the early detection of faults in railway axles is crucial in railway safety [1,2].

The ultrasound technique can be used to perform condition monitoring over railway axles; however, its disadvantage is that it does not provide continuous information between different tests; therefore,

it is not possible to detect fast-growing faults [3]. Other techniques that involve variables, such as temperature [4], acoustics [5], and acoustic emission [1], have been used for the continuous monitoring of faults on railway axles. However, vibration signal monitoring has become the most common monitoring technique due to its high reliability. Fault diagnosis based on vibration signals enables early fault detection, online condition monitoring, and when combined with different signal processing methods and artificial intelligence, better diagnostic results are obtained [6–8].

Fault diagnosis based on vibration signals with a data-driven approach is generally accomplished in four phases: (a) acquisition and conditioning of the vibration signal, (b) extraction of features, also called condition indicators, (c) selection of features, and (d) classification. After acquisition and conditioning of the vibration signal, in the feature extraction phase, we study the change of the signal behavior that can be an indicator of the fault condition. The study of these changes in the signal can be focused on the time, frequency or time-frequency domains. From these domains, condition indicators (CIs) or features can be extracted, which allows monitoring or detecting different faults [9]. The evaluation of the CIs has been widely studied, leading to achieve good results in the detection of faults in the railway axles. In the time-frequency domain, the energy calculated by means of Wavelet Packet Transform (WPT) has been used, allowing crack detection with excellent results [10,11]. The above-mentioned works only measured vibrations in the railway axle and bearings in insulation, without the bogie.

The work developed by Gómez et al. [7] (which is the same case of study of the present work), used real railway axles installed in a real Y21 bogie, where the vibration signals from six accelerometers were processed by means of the WPT energy. The feature selection stage was carried out by means of a visual analysis, the energy packages were selected to increase their values with the depth of the crack, and the packages varied with the change of speed. In the classification stage, a radial-basis function-based artificial neural network was used with 32 inputs corresponding to the selected energy packets, the load and speed values; the two possible outputs of the network were healthy or cracked conditions. This work highlighted that the six accelerometers provide important information for detection and better results are achieved at certain speeds. A recent work, over this same case, presented by Lucero at al. [12] evaluated the signals from the six accelerometers; thirty features from the time-domain signal were extracted, then, feature selection was applied, and finally the classification was implemented through a random forest classifier. The best accuracy in the classification was found with ten features, extracted from vibration signals measured with the accelerometers located in the longitudinal direction. The results in this work also show that features such as Wilson Amplitude (WAMP), Wave length (WL), Zero crossing (ZC), Slope Sign Change (SSC), mean, Energy Operator (EO), and the skewness are well-suitable to handle the fault classification. On the other hand, vibrations signals by nature exhibit random behavior in a wide range of applications. To reveal the strengths of different signal domains, the Fast Fourier Transform can be used to switch from time to frequency domain, and it has been noticed changes in the vibration signature of railway elements such as axle-boxes [13]. Moreover, if we want to understand how the strength of a signal is distributed in the frequency domain, we can use the power spectral density (PSD), which describes the power of the signal as a function per frequency unit. Therefore, the PSD can be used to infer normal operation or fault conditions of railway vehicle [14,15].

Through an analysis of CIs, it is possible to obtain adequate information for the understanding and interpretation of the machinery condition, such as the definition of limit values of certain indicators to establish that the machinery is in normal or abnormal conditions. This would support the diagnosis process and the maintenance decision-making [16,17]. On the other hand, the result of the diagnosis from the analysis of CIs can be improved by having several sensors to monitor the machinery because it would allow performing Data -Fusion [18,19]. The extraction of indicators in time and frequency domains requires a lower computational cost than the required one for calculating indicators in the time-frequency domain [20].

Data Fusion refers to the combination of data from multiple sensors of either the same or different types, and can be defined as the use of techniques that combine data from multiple sources. Thus, by using data fusion, a more reliable and realistic inference, deduction or discrimination can be made by using data from different isolated sources in data-driven approaches [21].

The goal of this work is the evaluation of the performance of condition indicators in time and frequency domains for crack detection in railway axles through the use of vibration signals. Two approaches are proposed: (1) the first one evaluates the indicators extracted from vibration signal of two accelerometers, in time domain, frequency domain, and their combination, and (2) the second approach evaluates the data fusion of the indicators of the six accelerometers. The rest of this paper is organized as follows. Section 2 presents the condition indicators used in this work, the selection, and classification methods. Section 3 contains the experimental set-up and data acquisition. In Section 4, the proposed methodology for evaluating the condition indicators performance is detailed. Then, Section 5 shows the results and discussion, and finally in Section 6, the conclusions are addressed.

## 2. Background

### 2.1. Condition Indicators

The analysis of signals can be done with several techniques and the use of signals on different domains can be useful to enrich the information obtained from a signal, leading us to a better understanding of its nature. If we are interested in quantifying some signal properties, we can use mathematics, statistics-based values or condition indicators to measure the different signal characteristics; this can help revealing the hidden information inside the signal. These condition indicators (CIs) are often called *features*. In the present work, the approach used in this feature extraction phase is a combination of different features computed from the signal in both time and frequency domains.

In case of time domain, we used 30 statistical indicators resulting from a mixture of common features mainly used in fault diagnosis and features coming from the Electromyography (EMG) field, such as described in [9]. Additionally, we used frequency-based CIs due to the fact that the frequency domain can reveal valuable information about the change in the monitored system condition in a different shape.

The frequency spectrum $X(k)$ of a discrete-time signal $x(i)$ can be computed by using the Fast Fourier Transform (FFT). The spectral analysis shows all the harmonic components of a signal, leading to a better understanding of the underlying phenomenon behavior. Another common approach to search signal characteristics in the frequency domain is the Power Spectral Density (PSD). The power spectrum of a signal $P(k)$ is given by

$$P(k) = \left(\frac{1}{K}\right)|X(k)|^2 \tag{1}$$

where $X(k)$ is a previously obtained frequency spectrum and $K$ denotes the number of points in the power spectrum. The PSD measures the average power of a signal in terms of the frequency, and it also shows periodicities [22,23]. The knowledge about the power distribution among the frequency components contained in a signal is also useful to understand the signal nature.

In frequency domain, we have used 24 condition indicators: 15 of them computed over the frequency spectrum and nine over the power spectrum. These 24 condition indicators are presented in Table 1; here, $f_k$ is the frequency value of the spectrum in the corresponding frequency bin $k$, whereas $K$ denotes the total number of samples in the frequency and power spectrum.

**Table 1.** Features.

| | |
|---|---|
| Mean of spectrum | $F1 = \frac{\sum_{k=1}^{K} X(k)}{K}$ |
| Variance of spectrum | $F2 = \frac{\sum_{k=1}^{K} (X(k)-F1)^2}{K-1}$ |
| Skewness of spectrum (Skewnessf) | $F3 = \frac{\sum_{k=1}^{K}}{(X(k)-F1)^3} K(\sqrt{F2})^3$ |
| Kurtosis of spectrum | $F4 = \frac{\sum_{k=1}^{K} (X(k)-F1)^4}{K(F2)^4}$ |
| Central Frequency | $F5 = \frac{\sum_{k=1}^{K} f_k X(k)}{\sum_{k=1}^{K} X(k)}$ |
| STD of spectrum | $F6 = \sqrt{\frac{\sum_{k=1}^{K}}{(f_k-F5)^2 X(k)} \sum_{k=1}^{K} X(k)}$ |
| RMS of spectrum | $F7 = \sqrt{\frac{\sum_{k=1}^{K} f_k^2 X(k)}{X(k)}}$ |
| CP1 | $F8 = \frac{\sum_{k=1}^{K} (f_k-F5)^3 X(k)}{K(F6)^3}$ |
| CP2 | $F9 = \frac{F6}{F5}$ |
| CP3 | $F10 = \frac{\sum_{k=1}^{K} (f_k-F5)^{\frac{1}{2}} X(k)}{K\sqrt{F6}}$ |
| CP4 | $F11 = \frac{\sum_{k=1}^{K} (f_k-F5)^3 X(k)}{F6^2 K}$ |
| CP5 | $F12 = \sqrt{\frac{\sum_{k=1}^{K} f_k^4 X(k)}{\sum_{k=1}^{K} f_k^2 X(k)}}$ |
| Centroid of Spectrum | $F13 = \frac{\sum_{k=1}^{K} k X(k)}{\sum_{k=1}^{K} X(k)}$ |
| Spectrum Spread | $F14 = \sqrt{\frac{\sum_{k=1}^{K} (k-F13)^2 X(k)}{\sum_{k=1}^{K} X(k)}}$ |
| Entropy of spectrum | $F15 = -\sum_{k=1}^{K-1} P_n(k) log_2 [P_n(k)]$ where $P_n$ is the normalized total spectral energy $P_n(k) = \frac{X(k)}{\sum_{k=1}^{K} X(k)}$ |
| Total power | $F16 = \sum_{k=1}^{K} P(k)$ |
| Median Frequency (MDF) | $F17 = \frac{1}{2} \sum_{k=1}^{K} P(k)$ |
| Peak frequency (PKF) | $F18 = max(P(k)), \quad k = 1, ... K.$ |
| First Spectral Moment | $F19 = \sum_{k=1}^{K} P(k) f_k$ |
| Second Spectral Moment | $F20 = \sum_{k=1}^{K} P(k) f_k^2$ |
| Third Spectral Moment | $F21 = \sum_{k=1}^{K} P(k) f_k^3$ |
| Fourth Spectral Moment (SM4) | $F22 = \sum_{k=1}^{K} P(k) f_k^4$ |
| Spectral moment ratio (VCF) | $F23 = \frac{F20}{F16} - (\frac{F19}{F16})^2$ |
| Frequency ratio (FR) | $F24 = \sum_{LLC=f_{min}}^{ULC=f_{max}/2} P(k) / \sum_{LHC=\frac{f_{max}}{2}+1}^{UHC=f_{max}} P(k)$ |

## 2.2. Random Forest for Feature Selection

Random Forest (RF) is a machine learning algorithm designed by Breiman in 2001 [24] for classification and regression. RF uses multiple decision trees to classify a sample; each decision tree is built by using bootstrap sampling with a random feature selection implementation. Their predictions are used in a voting

system where, from all trees, a majority class is calculated. The out-of-bag (OOB) error is mostly used to compute the expected model generalization performance [25].

In Random Forest, two methods defined by Breiman can be used for feature selection through of a feature ranking (feature importance): Mean Decrease Impurity (MDI) and Mean Decrease Accuracy (MDA). In this work, MDA is used for feature selection because if a feature does not impact the model, the method permutes the features values, such that the prediction accuracy should not decrease over the OBB observations.

Feature Selection by MDA

Given a data set $\mathbf{D_n} = \{(\mathbf{X_1}, Y_1), \ldots, (\mathbf{X_n}, Y_n)\}$ of $n$ samples and $p$ independent variables with $\mathbf{X}_i = (X_i^{(1)}, \ldots, X_i^{(p)})(i \in 1, \ldots, n)$ being a training sample, the importance of the j-th feature $\mathbf{X}^{(j)} = X_1^{(j)}, \ldots, X_n^{(j)} (j \in 1, \ldots, p)$, is calculated averaging the OBB errors of all permutations of trees [25].

Denote $D_{l,n}$ the out-of-bag data set of the $l-th$ tree and $D_{l,n}^j$ the same data set, where the values of $X^{(j)}$ have been randomly permuted. Keeping in mind that the $m_n(\cdot; \Theta_l)$ represents the $l$-th tree estimate and where $\Theta_1, \ldots, \Theta_M$ are independent random variable used to resample the training set before the growth of individual trees, MDA takes the form

$$MDA(X^{(j)}) = \frac{1}{M} \sum_{l=1}^{M} \left\{ R_n[m_n(\cdot; \Theta_l), D_{l,n}^j] - R_n[m_n(\cdot; \Theta_l), D_{l,n}] \right\} \tag{2}$$

where $R_n$ is defined for $D = D_{l,n}$ or $D = D_{l,n}^j$ by

$$R_n[m_n(\cdot; \Theta_l), D] = \frac{1}{|D|} \sum_{i:(\mathbf{X_i}, Y_i) \in D} [Y_i - m_n(\mathbf{X_i}; \Theta_l)]^2 \tag{3}$$

### 2.3. k-Nearest Neighbor Classifier

The K-Nearest Neighbor classifier (KNN) is a popular algorithm directly based on the training samples and commonly used in pattern classification [26].

To classify an unknown sample, two steps are followed. First, KNN calculates the distance between the unknown point $\mathbf{q}$ and points $\mathbf{x}_i$ in the training data, according to a distance metric $d(\mathbf{q}, \mathbf{x}_i)$. Generally, $d(\cdot, \cdot)$ can be Euclidean, Manhattan, Minkowski, Cosine, Chebychev Euclidean, Mahalanobis Standardized Euclidean, Hassana, or Chi-Square distance [27,28], just to mention a few distance metrics. Second, the $k$ nearest neighbors are used to determine the class of $\mathbf{q}$. The specified distance rule for classification of a new sample was Simple Voting. Then, an approach where new observation is assigned to the class of the majority of the k nearest points was used [29].

## 3. Experimental Set-Up

### 3.1. Bogie Test Bench

The bogie test bench was designed and manufactured by Danobat Railway Systems. It allows simulating different faults in the elements of the bogie. Figure 1 shows the main parts of the bogie, which is composed by the fixed wheels set (1) resting on the structure anchored to the floor (3). Wheels are connected by the fixed shaft (2), whereas the set of rotating wheels (6) are connected by the rotary axis (7); here is where the faults are simulated. The rotating wheels are driven by the rollers (10), and the speed is controlled by the roll driver (4), which is operated manually. The load is simulated by two hydraulic cylinders (9), and transmitted through a chain that pushes a beam (12) against the bogie structure (5).

Three accelerometers are located on the left side (11) and right side (8) to measure acceleration in all three directions on each side.

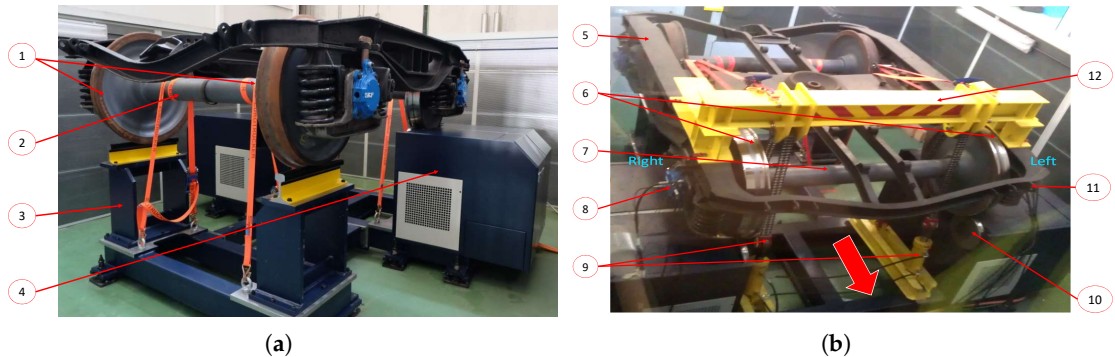

(**a**)    (**b**)

**Figure 1.** Components of the bed test. (**a**) Back view of the test bed. (**b**) Front view of the test bed.

### 3.2. Signal Acquisition and Experimental Conditions

A pair of bearings is included inside each axle box, supporting the rotating wheels. Three uni-axial accelerometers were placed at each axle box of the wheelset as indicated in Figure 2a, three on the right side (RS) and three on the left side (LS) oriented in three directions: left vertical (LV) accelerometer, left longitudinal (LL) accelerometer, and left axial (LA) accelerometer; right vertical (RV) accelerometer, right longitudinal (RL) accelerometer, and right axial (RA) accelerometer, as presented in Figure 2b. The accelerometer model is CMSS-RAIL-9100 with sensitivity of 100 mV/g, frequency range 0.52 Hz–8 kHz and resonance frequency 25 kHz, the accelerometer is coupled to the conditioner system SKF Multilog IMX-R, that was connected to a computer with the software SKF @ptitude Observer. This conditioner system is for industrial use and reduces the possibility of the signals having noise or interference. The sampling frequency was 12.8 kHz for a time of 1.2 s. The experimental conditions for load, speed, the rotation direction, and crack depths in the railway axle (see Figure 2) are presented in Table 2 for the four experimental conditions; at least 60 samples were acquired for each condition. We named the data obtained from each accelerometer: right-hand accelerometer data set (RD), and for the left and accelerometer data set (LD). Crack faults were artificially generated by an abrasive grinding process. Further details of the experimental conditions can be found in [7].

**Table 2.** Experimental conditions.

| | |
|---|---|
| 3 loads | - 4, 10, y 16 tons |
| 2 speeds conditions | - 20 km/h<br>- 50 km/h |
| 6 vibration measurement placements | - Left vertical (LV) accelerometer, left longitudinal (LL) accelerometer, left axial (LA) accelerometer<br>- Right vertical (RV) accelerometer, right longitudinal (RL) accelerometer, right axial (RA) accelerometer |
| Rotational directions | - Clockwise and counterclockwise |
| 4 fault conditions, see Figure 2 | - Healthy (Normal)<br>- Crack level 1 (e = 5.7 mm)<br>- Crack level 2 (e = 10.9 mm)<br>- Crack level 3 (e = 15 mm) |

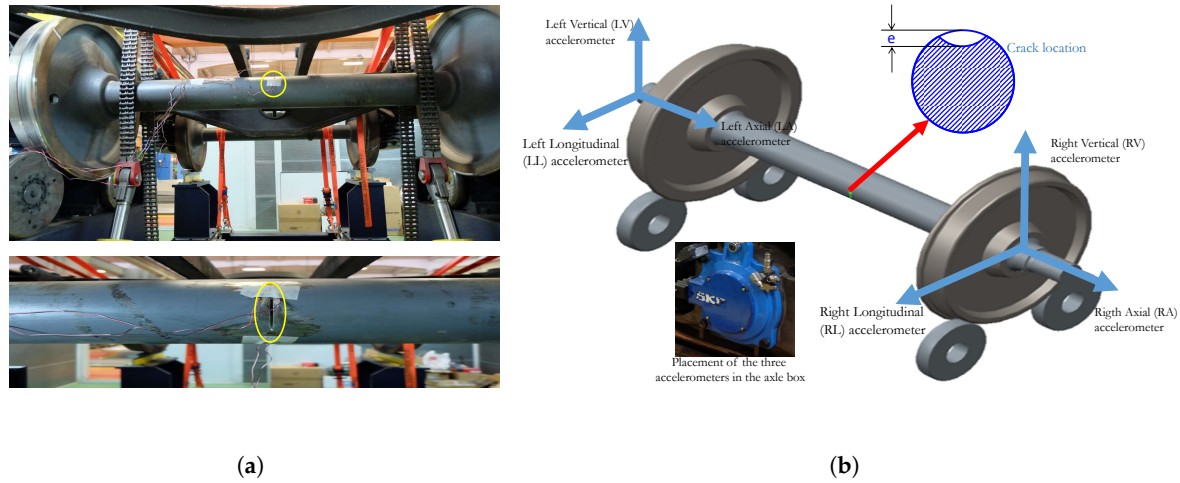

(**a**)                               (**b**)

**Figure 2.** Location of accelerometers and crack fault. (**a**) Location of the crack in the railway axle. (**b**) Location of accelerometers and crack fault.

## 4. The Proposed Approachs

This work proposes evaluating CIs for crack detection in railway axles. Two approaches are developed: the first approach evaluates the CIs of two accelerometers in the time domain, frequency domain, and combining them, the second approach evaluates the fusion of the CIs of all six accelerometers.

### 4.1. Proposed Approach 1

This approach evaluates condition indicators extracted from the vibration signal in time domain, frequency domain, and a combination of both called time + frequency domain for crack detection in railway axles. In this approach, we analyze the two accelerometers located along the longitudinal direction: on the right side (RL) and another on the left side (LL). According to the work developed by Lucero et al. [12], these accelerometers and their orientation offer the best information.

Figure 3a shows the workflow of the first approach. It has four stages: (1) Data Acquisition, (2) Feature Extraction, (3) Feature Selection, and (4) Classification. Each stage includes the following tasks:

1. Data acquisition: two sets of signals are acquired, from the left accelerometer (LL) along the longitudinal direction and from the right accelerometer (RL) along the same direction under different fault, load, and speed conditions, as explained in Section 3.2.
2. Feature extraction: the feature extraction was applied as explained in Section 3.2. Figure 4 shows examples of vibration signals in time domain, and signals in the frequency domain using FFT and PSD from both the left and right accelerometers, for normal (healthy) condition and fault level 1. For each signal, different CIs were obtained according to three domains: (i) 30 CIs for time domain (CI_T), (ii) 24 CIs for frequency domain (CI_F), and (iii) 54 CIs from time + frequency domain (CI_TF). Therefore, six data sets were completed. Three data sets were obtained with the left-side accelerometer along the longitudinal direction (LL): the first CI_TL contains CIs extracted from the left side accelerometer and in time domain, the second CI_FL contains the CIs in frequency domain for the same accelerometer, and finally, the third CI_TFL has CIs in the time + frequency domain, also with the same accelerometer. Similarly, with the right-side accelerometer along the longitudinal direction (RL), we obtained other three data sets: CI_TR, CI_FR, and CI_TFR on time, frequency and time + frequency domain, respectively.

3.  Feature selection: this process starts by removing correlated features. Correlation coefficient value on 0.8–1 between two CIs suggests these two features are highly correlated [30]. In this study, a threshold value equal to or greater than 0.95 (95%) was selected to identify two highly-correlated features, in all data sets. This threshold was stated after obtaining good performance in classification. Next, a normalization process was applied by scaling characteristics between −1 and 1. Normalization can be applied to Normal distributed data, as well as to data with another type of distribution. This process helps KNN to give equal importance to all CIs. After the preprocessing step, a Random Forest model was implemented with 40 trees as the main parameter. MDA metric was used to select the ten most important CIs, and then, the selection of the ten best-ranked features from each dataset.

4.  Classification: each dataset is organized in a matrix of samples in the rows and CIs in the columns. Each pre-selected dataset (built from 10 ranked CIs) in stage 3 is classified using KNN (a value of k = 3 was chosen as the main parameter after testing different number of neighbors) and the cosine distance metric was used.

Usually, when using k-folds cross-validation, values of k = 5 or k = 10 are chosen due to the good results obtained empirically [31]. In our work, the best results were obtained for k = 5. Therefore, a five-fold cross-validation strategy was carried out; the average accuracy and standard deviation (*std*) on the cross-validation process were calculated from 5 runs.

Finally, many classifications were performed; these starts by using the first feature for a classification, next, the two first features are used for a new classification, and so on, until reaching the ten first features. The purpose is to analyze the contribution of each ranked feature.

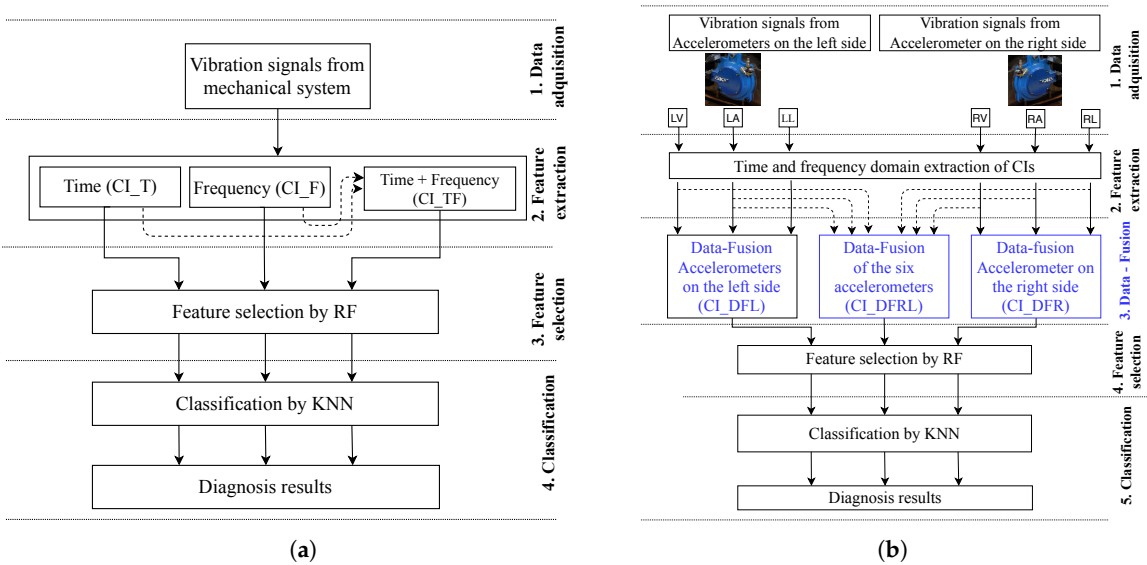

**Figure 3.** Flowcharts of the proposed approaches. (**a**) Flowchart approach 1 (**b**) Flowchart approach 2.

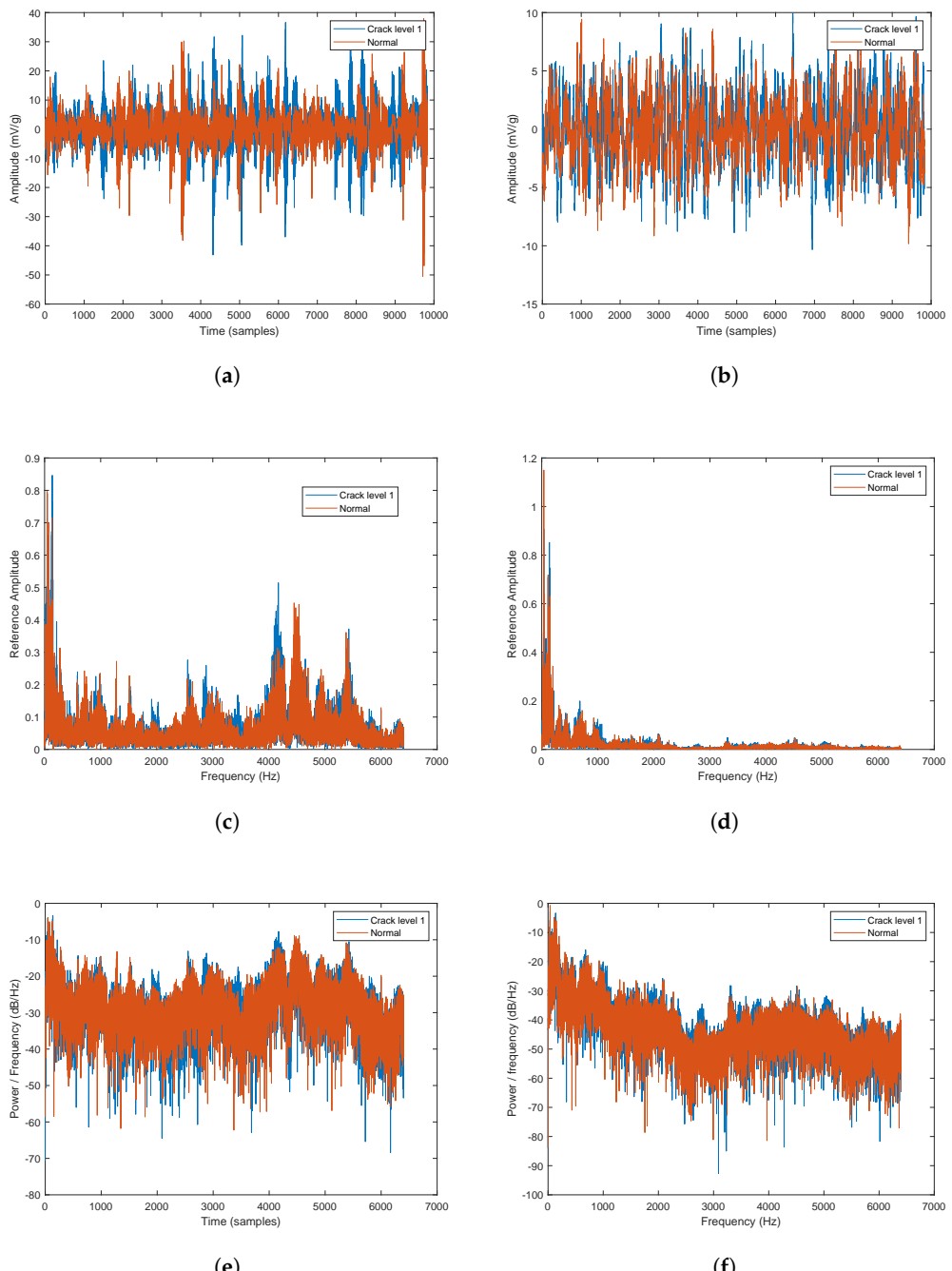

**Figure 4.** Sample vibration signal. Left and right longitudinal accelerometers with load of 4 tons, speed of 20 km/h, and crack level 1. (**a**) Time Signal left Accelerometer; (**b**) Time Signal right Accelerometer; (**c**) Frequency Spectrum left Accelerometer; (**d**) Frequency Spectrum right Accelerometer; (**e**) Power Density Spectrum left Accelerometer; (**f**) Power Density Spectrum right Accelerometer.

### 4.2. Proposed Approach 2

The second approach evaluates the signals from the six accelerometers using data fusion at the indicator level. Figure 3b shows the workflow of this approach. It has five stages: (1) Data acquisition, (2) Feature extraction, (3) Data-fusion, (4) Feature selection, and (5) Classification. Each of the stages is explained below.

1.  Data acquisition: the six vibration signals, three accelerometers mounted on the left side LV, LL, LA, and three accelerometers placed on the right side RV, RL, RA, were acquired under different faults, load and velocity conditions.
2.  Feature extraction: for each signal, different condition indicators were obtained according to two domains: (i) 30 CIs in time domain and (ii) 24 CIs in the frequency domain. All those condition indicators are combined, so, for each vibration signal, 54 CIs are obtained. Therefore, six data sets were used, three different sets of CIs per each accelerometer.
3.  Data-Fusion: from the six data sets of CIs obtained in the feature extraction stage, data fusion is performed at the indicator level. Three new sets of indicators are obtained: (i) The sets of CIs extracted from signals of the three accelerometers on the left side (LV, LA, LL) are fused, obtaining the first set of fused CIs called CI_DFL; (ii) the CIs of the three accelerometers on the right side (RV, RA, RL) are fused, obtaining a fused data set called CI_DFR; and finally (iii) the CIs of the six accelerometers are fused, obtaining the third set of fused CIs called CI_DFRL.
4.  Feature selection and classification: from the three data sets fused in stage 3, the next stages 4 (feature selection) and 5 (classification) were carried out by following the same procedures, experimental conditions, and CIs analysis of the stages Feature Selection and Classification of the approach 1 detailed in Section 4.1 and presented in Figure 3a.

## 5. Results and Discussion

### 5.1. Results of Approach 1

Tables 3–8 show the approaches to the results of approach 1. The top 10 CIs and their averaged accuracies for each data set are shown in Tables 3, 5 and 7. Tables 4, 6 and 8 show the accuracies per class for CI_TL, CI_FL and CI_TFL, respectively.

In Table 3, the best accuracy of 92.89%, was obtained by using **CI_TL** with 7 CIs. Otherwise, to **CI_TR** a maximum accuracy of 95.17% was obtained with 8 CIs. Per-class accuracy values for CI_TL are presented in Table 4. The best performance with 10 attributes shows that class 1 has the highest classification accuracy over 98% followed by class 2 with accuracy over 93%. Classes 3 and 4 have similar classification accuracy over 86%. Class 3 does not reach accuracy over 89% in any case. Class 4 can reach accuracy over 92% opposite to the accuracy of classes 1 and 2. Regarding the top 10 of time domain CIs presented in the Table 3, nine of the 10 are common: zero crossing, wave length, SSC, WAMP, kurtosis, energy operator, CPT3, shape factor, and skewness.

In Table 5, the best accuracy of 91.29% was obtained using **CI_FL** with 7 CIs. By contrast, to **CI_FR** a maximum accuracy of 94.36% was obtained with 9 CIs. Per-class accuracy values for CI_FL are presented in Table 6. The best performance with 10 attributes shows that class 4 has the highest classification accuracy over 98% followed by class 2 with accuracy over 89%. Classes 1 and 3 have similar classification accuracy over 85%. Classes 1, 2 and 3 do not reach accuracy over 89% in any case. Regarding the top 10 frequency domain CIs presented in the Table 5, seven of the 10 are common PKF, CP2, FR, skewnessf, centroid of spectrum, VCF, and spectrum spread.

In Table 7, the best accuracy of 96.62% due to its low std 0.59, was obtained using **CI_TFL** with 10 CIs. By contrast, to **CI_TFR** a maximum accuracy of 97.14% was obtained with also 10 CIs. Per-class values for

CI_TFL are presented in Table 8. The best performance with 10 attributes shows that classes 1 and 4 have similar high classification accuracy over 97%. Classes 2 and 3 have also similar classification accuracy over 95%. From the use of seven attributes, all the classes reach classification accuracy over 90%. Regarding the top 10 time + frequency domain CIs presented in the Table 7, six of the 10 are common; four belong to the frequency domain: PKF, CP2, FR, SM4; and two to the time domain: zero crossing and SCC. This result shows that the use of combined features from time and frequency domains can improve the classification accuracy for each class.

**Table 3.** Top 10 CIs and accuracies average over time domain.

| | CI_TL | | | | CL_RT | | |
|---|---|---|---|---|---|---|---|
| # | CIs | Accuracy (%) | Std | # | CIs | Accuracy (%) | Std |
| 1 | Zero crossing | 25.6 | 0.20 | 1 | SSC | 25.85 | 0.26 |
| 2 | Wave length | 60.45 | 3.26 | 2 | SRAV | 54.74 | 3.63 |
| 3 | Kurtosis | 76.63 | 2.04 | 3 | Wave length | 85.43 | 3.15 |
| 4 | WAMP | 85.07 | 1.29 | 4 | WAMP | 87.21 | 1.32 |
| 5 | SSC | 86.93 | 1.50 | 5 | Shape factor | 89.27 | 1.83 |
| 6 | CPT4 | 89.69 | 1.12 | 6 | Skewness | 90.43 | 1.85 |
| 7 | Skewness | **92.89** | 1.45 | 7 | Zero crossing | 93.65 | 2.46 |
| 8 | Shape factor | 92.89 | 1.18 | 8 | Kurtosis | **95.17** | 1.30 |
| 9 | CPT3 | 92 | 1.61 | 9 | Energy operator | 94.64 | 0.93 |
| 10 | Energy operator | 91.64 | 0.74 | 10 | CPT3 | 94.36 | 1.38 |

**Table 4.** Accuracy per class to CI_TL (time domain).

| # | CIs | Classes (Accuracy%) | | | |
|---|---|---|---|---|---|
| | | 1 | 2 | 3 | 4 |
| 1 | Zero crossing | 100 | 0 | 0 | 0 |
| 2 | Wave length | 65.95 | 55.44 | 48.58 | 71.92 |
| 3 | Kurtosis | 86.47 | 74.04 | 71.58 | 74.06 |
| 4 | WAMP | 89.23 | 84.56 | 83.09 | 83.21 |
| 5 | SSC | 90.97 | 84.56 | 86.34 | 85.76 |
| 6 | CPT4 | 94.45 | 88.77 | 84.55 | 90.9 |
| 7 | Skewness | 97.57 | 92.63 | 88.84 | 92.32 |
| 8 | Shape factor | 98.25 | 94.04 | 87.75 | 91.23 |
| 9 | CPT3 | 97.57 | 93.68 | 88.15 | 88.32 |
| 10 | Energy operator | 98.26 | 93.68 | 86.33 | 87.95 |

To evaluate the CIs in time, frequency and time + frequency domain, we compare the accuracy of each ranked feature set. Figure 5 shows the accuracy results for each data set. Here, the X-axis represents the number of $k$ first CIs most important selected by RF. The Y-axis represents the average accuracy of the $k$ first CIs.

In order to illustrate the importance of feature selection, a scatter plot obtained from the best case, related to the CI_FL set, is presented in Figure 6, where the class 1 is the red color, class 2 is the green color, class 3 is the cyan color, and class 4 is the violet color. This set provides the high accuracy and low *std* using 3 attributes. The formation of small clusters can be observed for each of the classes in different locations in the three-dimensional space, and no perfect boundaries are exhibited. However, these three attributes can provide over 80% of accuracy, according to Figure 5.

**Table 5.** Top 10 features and accuracies average over frequency domain.

| | CI_FL | | | | CI_FR | | |
|---|---|---|---|---|---|---|---|
| # | CIs | Accuracy (%) | Std | # | CIs | Accuracy (%) | Std |
| 1 | PKF | 25.6 | 0.15 | 1 | Skewnessf | 25.85 | 0.17 |
| 2 | CP2 | 56.8 | 2.93 | 2 | Spectrum spread | 41.77 | 2.44 |
| 3 | SM4 | 85.51 | 1.35 | 3 | PKF | 71.38 | 2.65 |
| 4 | FR | 72 | 2.54 | 4 | CP2 | 82.75 | 3.13 |
| 5 | Skewnessf | 79.03 | 2.57 | 5 | MDF | 83.99 | 2.48 |
| 6 | Spectrum spread | 87.21 | 2.43 | 6 | FR | 89.36 | 1.58 |
| 7 | VCF | **91.29** | 2.92 | 7 | CP3 | 94.01 | 1.16 |
| 8 | Centroid of Spectrum | 90.4 | 2.82 | 8 | SM1 | 93.47 | 1.32 |
| 9 | Kurtosisf | 90.13 | 1.08 | 9 | VCF | **94.36** | 0.93 |
| 10 | Entropy of spectrum | 90.23 | 1.97 | 10 | Centroid of Spectrum | 94.27 | 1.70 |

**Table 6.** Accuracy per class to CI_FL (frequency domain).

| # | CIs | Classes (Accuracy%) | | | |
|---|---|---|---|---|---|
| | | 1 | 2 | 3 | 4 |
| 1 | PKF | 100 | 0 | 0 | 0 |
| 2 | CP2 | 53.8 | 65.61 | 56.14 | 51.47 |
| 3 | SM4 | 78.14 | 85.96 | 84.53 | 93.8 |
| 4 | FR | 71.86 | 68.77 | 68.33 | 79.16 |
| 5 | Skewnessf | 81.98 | 77.89 | 74.79 | 81.41 |
| 6 | Spectrum spread | 85.4 | 83.86 | 84.18 | 95.62 |
| 7 | VCF | 89.24 | 89.82 | 87.77 | 98.54 |
| 8 | Centroid of Spectrum | 87.86 | 89.82 | 86.32 | 97.81 |
| 9 | Kurtosisf | 87.15 | 90.88 | 84.86 | 97.81 |
| 10 | Entropy of spectrum | 87.85 | 89.47 | 85.63 | 98.17 |

**Table 7.** Top 10 CIs and accuracies average over time and frequency domain.

| | CI_TFL | | | | | CI_TFR | | | |
|---|---|---|---|---|---|---|---|---|---|
| # | CIs | Domain | Accuracy (%) | Std | # | CIs | Domain | Accuracy (%) | Std |
| 1 | Zero crossing | Time | 25.6 | 0.15 | 1 | SSC | Time | 25.85 | 0.17 |
| 2 | PKF | Freq | 52.8 | 1.33 | 2 | FR | Freq | 50.81 | 1.77 |
| 3 | WAMP | Time | 73.43 | 3.37 | 3 | SM1 | Freq | 71.56 | 1.86 |
| 4 | CP2 | Freq | 78.85 | 1.91 | 4 | Skewnessf | Freq | 88.56 | 1.88 |
| 5 | SSC | Time | 82.49 | 1.89 | 5 | Spectrum spread | Freq | 93.83 | 1.67 |
| 6 | Shape factor | Time | 92.62 | 1.70 | 6 | Zero crossing | Time | 95.26 | 1.19 |
| 7 | Skewness | Time | 93.69 | 2.08 | 7 | SM4 | Freq | 94.63 | 1.15 |
| 8 | Kurtosis | Time | 94.49 | 0.89 | 8 | CP3 | Freq | 96.87 | 0.55 |
| 9 | SM4 | Freq | 94.4 | 2.35 | 9 | CP2 | Freq | 96.6 | 2.46 |
| 10 | FR | Freq | **96.62** | 0.59 | 10 | PKF | Freq | **97.14** | 1.29 |

**Table 8.** Accuracy per class to CI_TFL (time and frequency domain).

| # | CIs | Domain | Classes (Accuracy%) | | | |
|---|---|---|---|---|---|---|
| | | | **1** | **2** | **3** | **4** |
| 1 | Zero crossing | Time | 100 | 0 | 0 | 0 |
| 2 | PKF | Freq | 63.91 | 43.51 | 41.36 | 62.4 |
| 3 | WAMP | Time | 79.89 | 64.21 | 61.16 | 88.69 |
| 4 | CP2 | Freq | 77.11 | 75.09 | 72.64 | 90.88 |
| 5 | SSC | Time | 84.06 | 79.3 | 76.63 | 90.15 |
| 6 | Shape factor | Time | 93.04 | 89.47 | 92.79 | 95.25 |
| 7 | Skewness | Time | 96.19 | 90.88 | 96.77 | 90.86 |
| 8 | Kurtosis | Time | 96.88 | 92.28 | 96.39 | 92.32 |
| 9 | SM4 | Freq | 96.88 | 91.58 | 97.48 | 91.59 |
| 10 | FR | Freq | 97.21 | 95.09 | 96.4 | 97.81 |

## 5.2. Results of Approach 2

The top 10 CIs and their accuracies are presented in Tables 9 and 10 by using a Data Fusion approach, for all three data sets. The set of fused CIs from the left-side accelerometers (LV, LA, and LL) named CI_DFL, the set of fused CIs from the right-side accelerometers (RV, RA, and RL) named CI_DFR as presented in Table 9, and the third set of fused CIs from all six accelerometers named CI_DFRL as presented in Table 10. Figure 5 shows the accuracy results for each data set. Here, the X-axis represents the number of $k$ first CIs most important selected by RF. The Y-axis represents the average accuracy of the $k$ first CIs. Regarding the 10 top CIs presented in the Tables 9 and 10, five of the 10 are common, four belong to the frequency domain, i.e., PKF, FSK, VCF, skewnessf, and one to the time domain, i.e. SCC.

The highest accuracy (acc) using 10 condition indicators in CI_DFL was 97.56% with a std of 0.88, for CI_DFR was 98.01% with std of 0.88, and for CI_DFRL was 98.37% with std of 0.76. Of the latter, its accuracy by class is presented in the Table 11, where the class with the highest accuracy was number 4 obtaining 100% accuracy with 9 CIs.

In order to illustrate the importance of feature selection and data fusion in the approach 2, a scatter plot obtained from the best case, related to the CI_DFRL set, is presented in Figure 7. This set provides the highest accuracy (98.37%) and lowest *std* (0.76) using 3 CIs. Best small clusters can be observed for each class in contrast to Figure 6. The accuracy using 3 CIs was of 88.24%. These features were **MDF**-Freq-LV, **VCF**-Freq-RL, and **CP2**-Freq-RA.

**Table 9.** Top 10 features and accuracies average over time and frequency domain using left and right sensors.

| | CI_DFL | | | | | CI_DFR | | | |
|---|---|---|---|---|---|---|---|---|---|
| # | CIs - Domain | Sensor | Acc(%) | Std | # | CIs - Domain | Sensor | Acc(%) | Std |
| 1 | CPT6 - Time | LL | 25.5 | 0.13 | 1 | SSC - Time | RA | 25.5 | 0.17 |
| 2 | PKF - Freq | LA | 62.21 | 3.45 | 2 | Skewnessf - Freq | RV | 46.11 | 1.45 |
| 3 | FR -Freq | LV | 84.72 | 1.72 | 3 | SSC - Time | RL | 69.43 | 2.58 |
| 4 | MDF - Freq | LV | 93.67 | 0.85 | 4 | Skewnessf - Freq | RL | 87.16 | 1.2 |
| 5 | PKF - Freq | LV | 94.75 | 0.69 | 5 | CP2 - Freq | RL | 93.67 | 1.85 |
| 6 | Kurtosis - Time | LL | 96.38 | 1.4 | 6 | PKF - Freq | RV | 93.76 | 1.55 |
| 7 | Skewnessf - Freq | LV | 96.47 | 1.26 | 7 | ShapeFactor - TIime | RA | 96.48 | 1.78 |
| 8 | SSC - Time | LV | 96.56 | 1.59 | 8 | MDF - Freq | RL | 96.65 | 1.22 |
| 9 | SM4 - Freq | LA | 97.2 | 0.92 | 9 | VCF - Freq | RL | 97.92 | 1.05 |
| 10 | VCF - Freq | LV | **97.56** | 0.88 | 10 | ZC - Time | RL | **98.01** | 0.88 |

**Table 10.** Top 10 features and accuracies average over time and frequency domain using all sensors, CI_DFRL.

| | | | CI_DFRL | | |
|---|---|---|---|---|---|
| # | CIs | Domain | Sensor | Accuracy (%) | Std |
| 1 | MDF | Freq | LV | 25.5 | 0.19 |
| 2 | VCF | Freq | RL | 63.47 | 3.96 |
| 3 | CP2 | Freq | RA | 88.24 | 1.12 |
| 4 | SSC | Time | LV | 92.68 | 1.67 |
| 5 | CP2 | Freq | LL | 94.21 | 1.87 |
| 6 | PKF | Freq | LV | 96.38 | 1.24 |
| 7 | PKF | Freq | RL | 97.02 | 0.82 |
| 8 | CP2 | Freq | RL | 97.2 | 1.14 |
| 9 | ZC | Time | LL | 97.38 | 0.38 |
| 10 | Skewnessf | Freq | LV | **98.37** | 0.76 |

**Table 11.** Accuracy per class to CI_DFRL.

| # | CIs | Domain | Sensor | Classes (Accuracy%) | | | |
|---|---|---|---|---|---|---|---|
| | | | | 1 | 2 | 3 | 4 |
| 1 | MDF | Freq | LV | 100 | 0 | 0 | 0 |
| 2 | VCF | Freq | RL | 70.94 | 46.94 | 25.82 | 39.88 |
| 3 | CP2 | Freq | RA | 84.4 | 64.75 | 66.18 | 61.97 |
| 4 | SSC | Time | LV | 90.76 | 84.72 | 79.64 | 93.65 |
| 5 | CP2 | Freq | LL | 96.46 | 94.29 | 85.82 | 98.13 |
| 6 | PKF | Freq | LV | 96.8 | 93.93 | 86.55 | 97.77 |
| 7 | PKF | Freq | RL | 97.53 | 96.44 | 93.09 | 98.88 |
| 8 | CP2 | Freq | RL | 96.83 | 96.09 | 94.18 | 99.62 |
| 9 | ZC | Time | LL | 97.88 | 97.51 | 96.36 | **100** |
| 10 | Skewnessf | Freq | LV | 98.58 | 97.86 | 96.73 | 98.88 |

*5.3. Discussion*

The proposed approaches provide the most efficient CIs to be evaluated for crack detection. The results in Tables 3, 5, 7, 9 and 10 show the approach 1 achieves an accuracy rate of more than 91% from the seventh CI for all data sets while approach 2 achieves a rate of more than 93% from the fifth CI. Furthermore, the *std* obtained by each K-fold is presented in these tables where it can be seen that new CIs help reducing the *std* of the cross-validation process. This result indicates the classifier becomes more robust to new features. Note that using the *std* metric from the cross-validation process is a more useful, unlike the typical single train/test experiment.

In Figure 5, the trend of the curves indicates a fast growth of the accuracy up to the seven CIs, later the value of the accuracy flattens, i.e. the classifier will not significantly improve the accuracy after adding more CIs. It is important to note that all trends present a continuous increment in the accuracy as CIs increases except for CI_FL, for which the fourth CI decreases its performance.

Figure 8 shows the maximum classification accuracy of each data set and the number of indicators for which it was obtained. Approach 1, which employs a single accelerometer, achieves the best results by combining the CIs of time + frequency domain for both the left and right side accelerometers. Slightly better classification results are achieved with approach 2; however, data fusion by using six-accelerometers may involve greater challenges to perform the system diagnostic. For the three domains analyzed, better results are achieved with the right-side accelerometers, which may be due to the fact that the drive is not on this side and the signal information is less noisy.

Regarding the other works in this same case, Lucero et al. [12] reached a maximum accuracy of 96.43% and Gomez et al. [7] achieved the overall probability of detection at 95%, with 32 features (energy computed on WPT). In this work, the maximum accuracy is 97.14% with the approach 1, and 98.37% with the approach 2.

The ZC and SCC condition indicators of time domain and PKF, CP2 in frequency domain, are the common CIs among the top 10 CIs in both approach 1 and 2.

One of the main benefits of approach 1 is the possibility of classifying cracks in railway axles with a few CIs of time + frequency domain extracted from signals of a single accelerometer. In this way, we can reduce the number of machine sensors to detect a crack. On the other hand, eliminating non-informative indicators improves the performance of the classification algorithm. In addition, the use of few features reduces the computational cost of data processing.

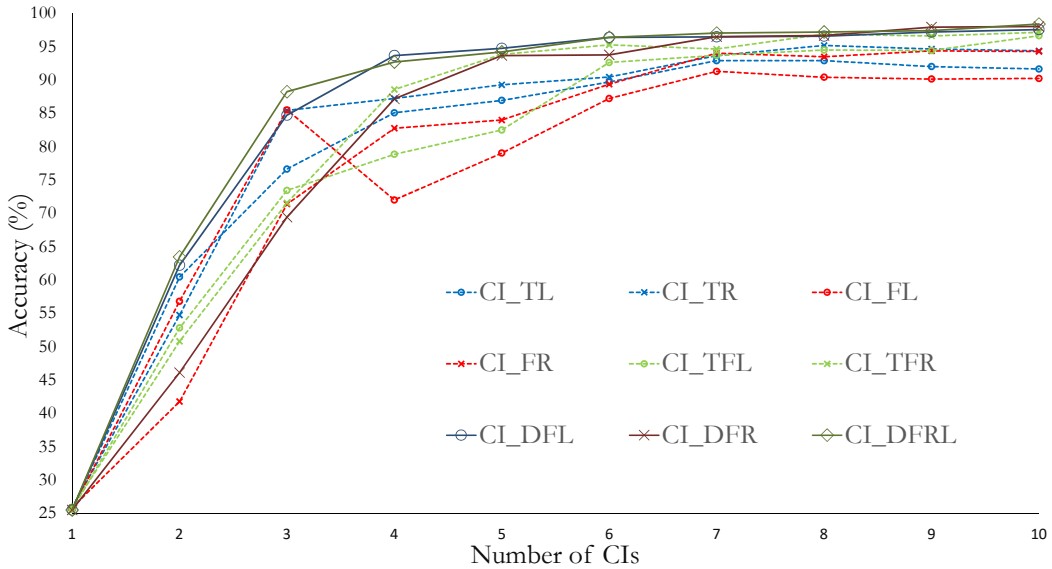

**Figure 5.** Accuracy vs. different numbers of selected CIs.

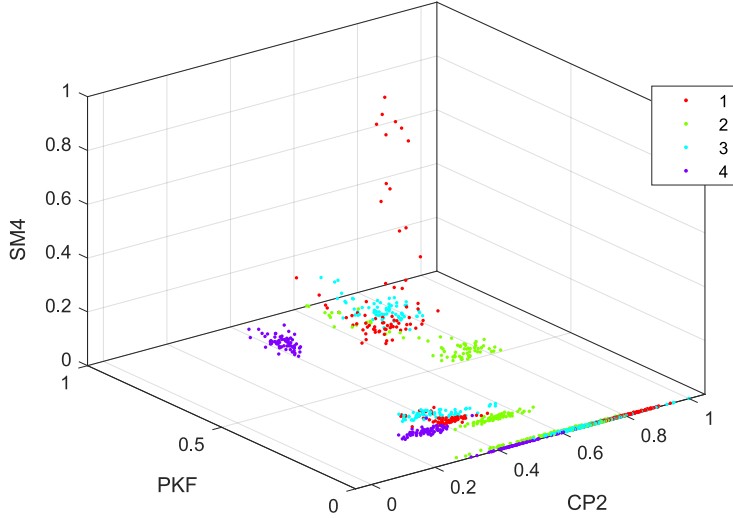

**Figure 6.** Scatter plot using 3 best CIs of CI_FL.

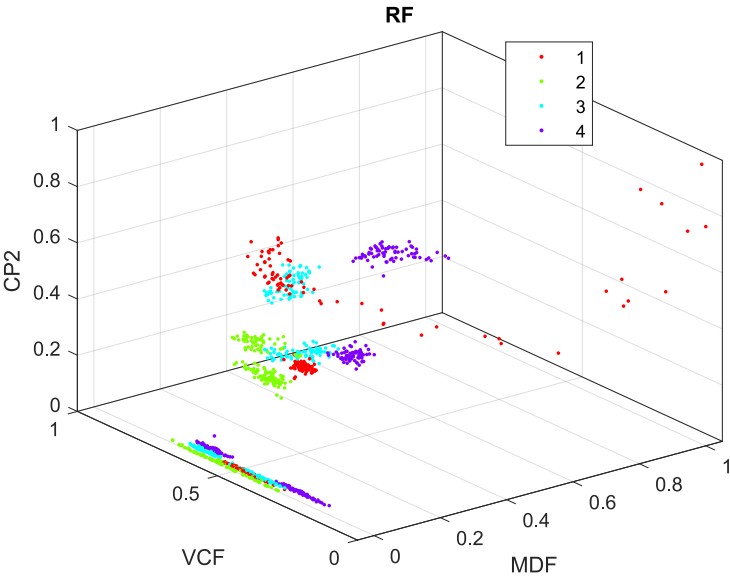

**Figure 7.** Scatter plot using 3 best CIs of CI_DFRL.

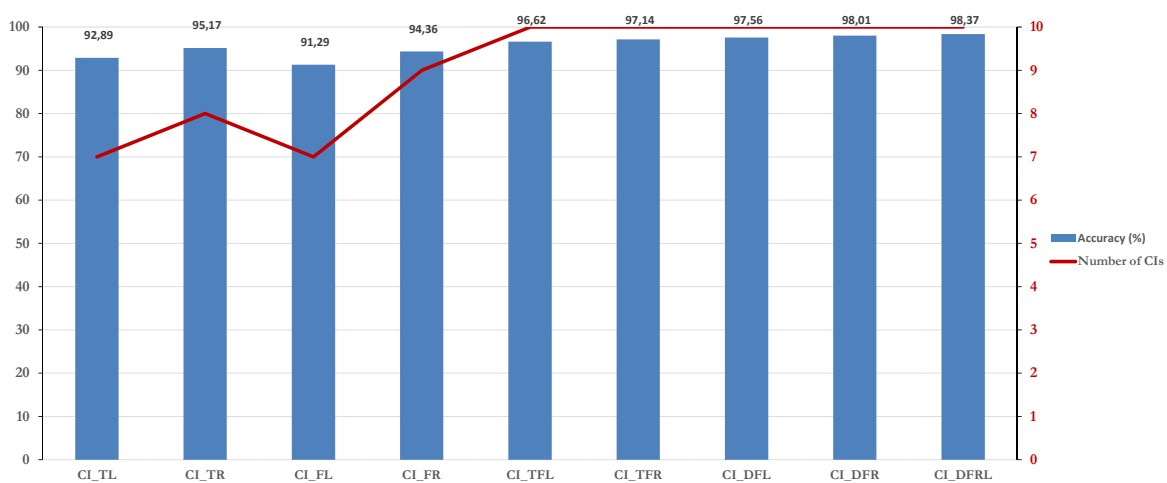

**Figure 8.** Highest classification accuracy achieved per number of CIs.

## 6. Conclusions

In this work, it was possible to develop a practical and straightforward methodology to evaluate the performance of condition indicators (CIs) in time and frequency domains. Its application was tested for crack detection in railway axles, by using vibration signals measured with accelerometers located on the right and left side of the railway axles, along the vertical, axial, and longitudinal directions. This test bed is made of real components and has been used in previous works for testing other approaches of vibration analysis.

Two approaches were proposed. Approach 1 only analyzes the CIs extracted from the vibrations signals measured with the accelerometers placed along the longitudinal direction, on the right and left sides of the axles, as an extension for comparing to previous works analyzing this case. Approach 2 uses a data fusion analysis, by combining the CIs extracted from all the six accelerometers.

In approach 1, CIs extracted from the accelerometer on the right side in the different domains show slightly better results of classification accuracy than the condition indicators from the accelerometer on the left side. The right side accelerometer time + frequency domain data set (CI_TFR) had the highest performance, with classification accuracy over 97.14%, which is better than the results obtained in previous works. This fact may be due because the drive motor is placed on the right side, and the driven motor is on the left side which provides movement to the rotating wheels.

In approach 2, the best performance was achieved with CIs in the data set CI_DFRL which combines all the indicators extracted from the six accelerometers. Classification accuracy was over 98.38%, and this result is better than those one by using approach 1, showing that Data Fusion is a good approach to improve accuracy in fault classification.

The best results in terms of classification accuracy were achieved by using a combination of indicators extracted from time and frequency domains. The CIs named ZC and SCC in the time domain and PKF, skewnessf, CP2 in the frequency domain, are the common CIs among the top 10 CIs in both approach 1 and approach 2.

The main advantage of this methodology is that it supports the identification of the relevant CIs for fault detection in different domains of the vibration signal. This allows identifying which CIs should be monitored on-line, mainly the CIs of time domain as they have low computational cost regarding the signal processing.

As future work, the contribution of each single attribute to improving the accuracy values regarding the crack detection in railway axles will be analyzed. Moreover, the analysis of CIs coming from other application domains will be developed. Additionally, it is expected to have a repository of discriminating and independent features to develop effective data-driven classification algorithms.

**Author Contributions:** R.-V.S. proposed the idea, H.R.A. and C.C. provided the experimental design and data, J.-C.M. implemented the data processing algorithms and wrote a draft of the paper, P.L. implemented classification algorithms and performed results analysis, and M.C. and D.C. supervised the work and reviewed the final manuscript. All authors contributed in the proposed methodology. All authors have read and agree to the published version of the manuscript.

**Funding:** This research received no external funding, the APC was funded by Universidad Politécnica Salesiana through the research group GIDTEC.

**Acknowledgments:** Authors would like to thank the support provided by the Spanish Government, through the MAQ-STATUS DPI2015-69325-C2-1-R project, and Universidad Politécnica Salesiana through the research group GIDTEC.

**Conflicts of Interest:** The authors declare no conflict of interest.

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
