# Peer review of "Evaluation of Time and Frequency Condition Indicators from Vibration Signals for Crack Detection in Railway Axles"

_applsci, doi:10.3390/app10124367_

Round 1

Reviewer 1 Report

The study is interesting, especially the part where authors use Random forests to select features. However, numerous flaws are spotted. Moreover, this manuscript has to be significantly improved.

1) Introduction and onwards - it is worth to mention that Power Spectral Density is appropriate to analyze random vibration signals and not periodic or transient ones.

2) Introduction section calls for more extensive literature review. This should focus on particular machine learning techniques and particular damage-sensitive features.

3) Section 2: Additionally, we used frequency-based CIs due to the fact that the frequency domain...

4) Section 3: Figure 1 shows the main parts of the bogie, which is comprises of the fixed...

5) Present some more data for accelerometers - sensitivity, usable frequency bandwidth, resonance frequency, etc.

6) No actual recorded signals in time domain and frequency domain are shown to compare signals from left and right accelerometer.

7) How do authors estimate the degree of correlation of 95 % to be a threshold for feature removal? Maybe 90-95  is more appropriate? What is the objective reasoning behind that?

8) Section 4, 4. Classification: a values of k=3 was chosen as the main parameter after testing different...

9) Section 4, 4. Classification: Finally, many classifications were performed; these start by using...

10) What is the nature of bogie excitation? The loading magnitudes are stated but is the excitation itself random, periodic or transient?

11) Section 5: To evaluate CIs in the time, frequency and time+frequency domain, we compare... (combine these 2 sentences).

12) No actual scatterplots are shown to see how classes are separated. What are the decision boundaries for kNN classifier?

13) Error analysis is weak - what are the cross-validation and resubstitution errors? Only accuracy is presented.

14) What distance metric was used for kNN? What distance weighting, if any, was used?

15) Was the data standardized before classification with kNN? It usually has to be standardized if different sensor data is to be fused together.

16) Which classes are misclassified? Need to compute the confusion matrices.

Reviewer 2 Report

This Paper discuss the performance of condition indicators in time and frequency domain for crack detection in railway axles through the use of vibration signals. This topic has been studied extensively by many authors and therefore the authors should state the novelity very clear in the introduction section. in addition the author should consider the following to improve the paper quality:

  1. in the introduction section the authors used the second pargraph to discuss the acoustic emissions, while this paper use only vibration analysis. So this paragraph is irrelvant.
  2. The background need significant improvement, the authors should emphasis more on the condition indicators and described in full and provide comperhensive litreature review on how this CI used for similar study on crack detection such as the papers below:
    1. https://doi.org/10.1109/SDPC.2017.112
    2. https://doi.org/10.1016/j.apacoust.2017.01.005
    3. https://doi.org/10.1016/j.matpr.2017.12.219
  3. in page 4, the authors has stated " In this work, we analyze the two accelerometers located in the longitudinal113 direction: ontherightside(ARS)andotherontheleftside(ALS" please justify the selection of this direction.
  4. Please provide more details about the fault size and location (may be additional figure)
  5. In result section, please provide the general trend of the condition indicators.
  6. Justify why some condition indicators provided better accuracy.

Round 2

Reviewer 1 Report

Not many improvements were spotted, although the Introductions section has been considerably improved.As it is now, the paper can't be accepted for publication. It must be thoroughly revised. The English also has to be double checked.

My comments:

1) Subsection 2.2: Brieman or Breiman?

2) Use of PSD is meaningful only in case of random signals, while FFT is used only for harmonic signals. What type of signals were measured and why responses in time domain and frequency domain are different for both sensors?

3) Subsection 2.3: There can be other distances besides the ones you mentioned. Mahalanobis distance is often used, for example.

4) Subsection 2.3: standardization of the data prior to kNN application is essential for the features to have comparable ranges. It was mentioned briefly in one place, but this definitely has to be more stressed out.

This brings to the next question - how distributed is the data and what is the final data size used in the classification? Does the data follow normal distribution?

5) Subsection 2.3: what about distance weighting? Did you use any?

6) The justification to consider only longitudinal direction for accelerometers just because the results were better in other studies is not acceptable. You should analyze all directions yourself and then conclude on which direction was more favorable in this regard.

7) What is the reason to select 10 best features and not any other number?

8) Was the raw measurement data in time domain pre-processed in some way (filtering,de-noising)? If not, there is a strong possibility that noise has had a negative influence on the results.

9) No frequency peaks are identified in the spectra, so it is difficult to judge if the sampling frequency could have been lower or not.

10) Overall organization of the data for the classification is missing. Usually it is displayed in matrix form with columns for features and rows for observations for every class label.

11) Figure 6: which classes correspond to 1, 2,3, 4? How good is the separability of the classes? Include the distances between classes for the distance metric you used in the study.

12) Figure 5: it is fine, but you could have introduced some cut-off beyond which the addition of new features is no longer feasible. The reason is that while with every new feature above the cut-off the accuracy improvements are very small, the computational time may increase more significantly. So I suggest to add a second y axis to Figure 5 to show computational time vs number of features.

13) Error analysis is lagging. The confusion matrix (CM) could have been computed to show how well the classification is done for every class separately. You can plot CM for both classes separately or combine results from both sensors into one and then try computing CM. Will there be a difference? True positives and negatives, as well as false positives and negatives should be computed and compared for both sensors and different features.

I would have liked to see a K-fold analysis, since you used K-fold CV. The number of K folds actually depends on the data size. The more data you have, the less folds are needed. Again, stating the data size is crucial.

Analysis of resubstitution loss analysis is also important and is missing in this work.

14) Conclusions: must be improved. No clear results stated in terms of numbers. Which features are better than others and by how much. What about error rates (see comment 13 - after you have calculated those quantities, include the most important ones in conclusions). Also, the limitations of the method must be discussed.

Reviewer 2 Report

the novelaty and contribution of this research still not clear, authors should discuss clearly what the contribution of this research.
